# Facile Synthesis of Bio-Antimicrobials with “Smart” Triiodides

**DOI:** 10.3390/molecules26123553

**Published:** 2021-06-10

**Authors:** Zehra Edis, Samir Haj Bloukh

**Affiliations:** 1Department of Pharmaceutical Sciences, College of Pharmacy and Health Science, Ajman University, P.O. Box 346 Ajman, United Arab Emirates; 2Center of Medical and Bio-Allied Health Sciences Research, Ajman University, P.O. Box 346 Ajman, United Arab Emirates; s.bloukh@ajman.ac.ae; 3Department of Clinical Sciences, College of Pharmacy and Health Science, Ajman University, P.O. Box 346 Ajman, United Arab Emirates

**Keywords:** iodine, triiodides, *Salva officinalis* L., *Aloe vera*, halogen bonding, antimicrobial resistance, biomaterials, antimicrobial activity, synergism, nosocomial infections

## Abstract

Multi-drug resistant pathogens are a rising danger for the future of mankind. Iodine (I_2_) is a centuries-old microbicide, but leads to skin discoloration, irritation, and uncontrolled iodine release. Plants rich in phytochemicals have a long history in basic health care. *Aloe Vera Barbadensis* Miller (AV) and *Salvia officinalis* L. (Sage) are effectively utilized against different ailments. Previously, we investigated the antimicrobial activities of smart triiodides and iodinated AV hybrids. In this work, we combined iodine with Sage extracts and pure AV gel with polyvinylpyrrolidone (PVP) as an encapsulating and stabilizing agent. Fourier transform infrared spectroscopy (FT-IR), Ultraviolet-visible spectroscopy (UV-Vis), Surface-Enhanced Raman Spectroscopy (SERS), microstructural analysis by scanning electron microscopy (SEM), energy dispersive spectroscopy (EDS), and X-Ray-Diffraction (XRD) analysis verified the composition of AV-PVP-Sage-I_2_. Antimicrobial properties were investigated by disc diffusion method against 10 reference microbial strains in comparison to gentamicin and nystatin. We impregnated surgical sutures with our biohybrid and tested their inhibitory effects. AV-PVP-Sage-I_2_ showed excellent to intermediate antimicrobial activity in discs and sutures. The iodine within the polymeric biomaterial AV-PVP-Sage-I_2_ and the synergistic action of the two plant extracts enhanced the microbial inhibition. Our compound has potential for use as an antifungal agent, disinfectant and coating material on sutures to prevent surgical site infections.

## 1. Introduction

Increasing resistance to multi-drug resistant pathogens against common antibiotics and antimicrobial agents endangers the survival of our species [1,2]. Nosocomial infections, steadily increasing treatment duration and costs slash quality of life throughout the world [1,2,3]. This trend quietly impacted COVID-19 related treatment outcomes and death rates behind the scenes [4,5]. Nosocomial infections of severely ill patients in health care settings exacerbated pain, suffering and death [5,6]. Combatting COVID-19 requires administering antibiotics to ameliorate the health condition of the patients [5,6]. Plant-based alternatives targeting bacterial virulence factors and biofilm formation emerge increasingly as sustainable solutions against microbial resistance [7,8]. Nano-biomaterials revealed promising results as antimicrobial agents either as plant nanoparticles or biosynthesized nanoparticles [8,9]. Silver nanoparticles are preferred for drug delivery and as antimicrobial agents, but may have possible cytotoxic properties and unknown longterm effects [10,11,12,13,14]. Plant and herbal extracts are proven to have medicinal uses due to their constituents like polyphenols, flavonoids, and further classes of phytochemicals [15,16,17]. Such plant-based antimicrobial agents offer valuable solutions even against the notorious ESKAPE (*Enterococcus faecium*, *Staphylococcus aureus*, *Klebsiella pneumoniae*, *Acinetobacter baumannii*, *Pseudomonas aeruginosa*, *Enterobacter* spp., and *Escherichia coli*) pathogens [2,17]. Plant species survived through naturally evolved synergistic mechanisms by a plethora of phytochemicals in the fight against microorganisms [15,16,17]. These constituents are an integral part of a holistic approach towards the phenomenon of resistance. Natural plant products in the form of herbs and spices have been known and utilized throughout the history of mankind against health problems. Herbs and spices are easily accessible for every household. Their low-cost and eco-friendly pharmacological components can be unlocked by the simplest preparation methods. They represent a chance for low-income, under-developed populations [14].

*Salvia officinalis* L. (Sage) has been a well-known remedy for different ailments in many cultures through the centuries [18,19]. The demand for new classes of antimicrobial agents has produced an increasing number of publications about antibacterial and antifungal properties of *Salvia officinalis* L. in recent years [14,15,16,17,18,19,20,21,22,23,24,25]. Sage can be used for biosynthesis of nanoparticles in anticancer, antimicrobial, antimalarial and other applications [14,20,21]. Sage is rich in the bioactive compounds flavonoids, terpenes and polyphenols in increasing order [18,19,20,21,22,23,24,25]. The polyphenols are mainly rosmarinic acid, caffeic acid, vanillic and ferulic acid, which can be extracted by different methods [22,23,24,25,26,27,28].

*Aloe Vera Barbadensis* Miller (AV) has been used for centuries against ailments due to its antimicrobial and anti-inflammatory properties [29,30]. AV consists of a plethora of different phytochemicals, which act by synergistic mechanisms against microbial pathogens [29,30,31,32,33,34,35,36,37,38,39,40,41]. AV has shown antiviral activities against different flu strains [32,33]. AV components like aloe-emodin prevent biofilm formation of multi-drug resistant *S. aureus* [34]. Acemannan, a major polymeric constituent of AV is a potent antimicrobial agent [35]. AV is effective against oral pathogens and can be used as an ointment and a wound dressing material [36,37,38,39,40]. The main active ingredients are acemannan, aloin, aloe-emodin and hesperidin, accompanied by further bioactive compounds [29,30,31]. The medicinal uses and antimicrobial activities of AV depend on factors like harvest season, climate, soil, irrigation pattern and extraction method [31,37,42,43,44,45]. AV and Sage are natural products, which may have variable composition according to their environmental factors during growth, harvesting and extraction methods [21,31,37,42,43,44,45]. These factors influence their biological activities and may be standardized by strictly controlling environmental factors and extraction methods. Synergetic mechanisms enhance their benefits, enabling their uses against a variety of conditions independent from such factors [7,15,16,17,23]. In our previous study, we enhanced the antimicrobial activities of AV by incorporating the antimicrobial agent iodine [31].

Iodine is a well-known microbicide and has been used traditionally in every culture as an antimicrobial agent despite its side effects of skin discoloration and irritation [46,47,48]. Another drawback of iodine is the uncontrolled and fast release, resulting in short-term activity [46,47,48]. These disadvantages can be controlled by stabilizing the iodine structure. Iodine can form polyiodides through attachment of iodide anions (I^−^), triiodide ions (I_3_^−^) and iodine molecules (I_2_) as the basic units [47,49,50]. The polyiodides can vary from isolated to two- and three-dimensional networks with polymeric structure [47,49,50,51]. Polyiodides contain different patterns of bonding, including halogen bonding [47,52,53,54,55,56,57]. Halogen bonding can enhance the stability of triiodides and enable controlled iodine release [56]. This allows use of iodine in lower concentrations and eradicates the disadvantages of skin discoloration, skin irritation and uncontrolled iodine release [46,47,48,56]. The polyiodide structure also strongly depends on the surrounding atoms and can be stabilized by complexation [58,59,60,61,62,63,64]. Crown ether molecules like 12-crown-4 render stable polyiodide-structures, which can be used for antimicrobial applications [59,60,61,62,63]. Polymer-iodine complexes are another alternative with many antimicrobial applications [47,64,65,66,67,68,69,70,71,72]. Povidone-iodine, a combination between iodine and polyvinylpyrrolidone (PVP), is an example of a popular microbicide with anti-pathogenic effects even against resistant microorganisms [47,66,67]. Complexed polyiodides can be used as coating agents, wound dressing materials and in combination with silver nanoparticles [67,68,69,70,71,72]. Coating surgical sutures with iodine-polymer complexes amplify inhibition of harmful pathogens on-site and can reduce the incident of surgical site infections [70,72]. Minimizing inflammation and biofilm formation will result in a faster wound healing process [65,69,70,73].

In our previous study, we prepared “smart” triiodides with halogen bonding within a PVP matrix by adding iodine (I_2_) and sodium iodide (NaI) to the AV-PVP hybrid [31]. In this work, we added the ethanolic maceration extract of fresh *Salvia officinalis* L. (Sage) plants into the bio-hybrid. The antimicrobial properties were enhanced through synergism of the biomaterials within AV and Sage. We used fresh Sage plants consisting of leaves and stems. These were extracted by solid-liquid extraction in ethanol. To our knowledge, we are the first to discuss the antimicrobial activity of a combination of pure AV gel with fresh Sage plant extract on this selection of 10 reference microbial strains.

We combined pure AV gel and different Sage extracts with iodine through a cost-effective and easy one-pot synthesis with polyvinylpyrrolidone (PVP) as encapsulating and stabilizing agent. Fourier transform infrared spectroscopy (FTIR), Ultraviolet-visible spectroscopy (UV-Vis), Surface-Enhanced Raman Spectroscopy (SERS), microstructural analysis by scanning electron microscopy (SEM), energy dispersive spectroscopy (EDS) and X-Ray-Diffraction (XRD) analysis verified the composition of AV-PVP-Sage-I_2_. The antimicrobial testing by disc diffusion method against 10 reference microbial strains showed excellent to intermediate antimicrobial activity in discs and sutures. We compared the results with the antibiotics gentamicin and nystatin. Our compound exhibited stronger antifungal activity than nystatin. The triiodide moieties within the biomaterial induced controlled release of iodine and resulted in enhanced microbial inhibition, enabling use for the prevention of surgical site infections in sutures and disinfectants.

## 2. Results and Discussion

Iodine, Sage, and AV have been known remedies throughout the history of mankind. The combination of these three components in one formulation may render promising microbicides. Our previous investigations on “smart” triiodides and iodinated biomaterials confirm this hypothesis [31,56,72]. Such triiodides with pure halogen bonding are stable and only release molecular iodine when the biopolymer structure is deformed by electrostatic interactions with cell membranes of pathogens [31,56,61]. The UV-vis spectroscopic studies confirmed the availability of “smart” triiodides in our title compound. These biohybrids consist of readily available ingredients for most households. Our aim is to offer basic health care solutions for the public, which can be reproduced and utilized by everyone easily. We believe that nature based, simple formulations may be an answer to the increasing threat of antimicrobial resistance and health care emergencies like future pandemics. Sage and AV contain a spectrum of bioactive compounds, which include phenolic acids, polyphenols, anthraquinones and further phytochemicals [18,19,29,30,31,32,33,34,35,36,37,38,39,40,41].

The antimicrobial activities of phenolic acids depend on their lipophilicity, number of hydroxyl-, ester- and methoxy-groups [23]. Hydroxyl-groups directly interact with the polar heads of the phospholipid components of cell membranes by hydrogen bonding and lead to membrane disruption by aggregation, reduced fluidity and increased rigidity [23]. The membrane disruption is accompanied by leakage of cell interiors and leads to cell death [23]. Phenolic acids like caffeic acid are lipohilic and pass through the cell membrane to the cytoplasm and acidify the cytosol [23]. Efflux pumps in bacterial strains can be disabled by phenolic acids [23]. Phenolic acids disrupt quorum-sensing communication and inhibit biofilm formation [15,23]. Extraction methods are one of the key factors of preserving the needed antimicrobial biocompounds in plants like Sage and AV. Extraction temperature, solvents and methods directly impact the resulting bioactivity of the products and need to be employed accordingly [18,19,20,21,22,23,24,25,26,35,37,45].

Maceration of Sage plants with methanol for 72 h renders ferulic acid, followed by rosmarinic acid and the flavonoid apigenin [18,25]. Aqueous, alcoholic extracts of Sage contain rosmarinic acid and luteolin-7-glucoside [19,26]. The extraction methods decoction, infusion, soxleth, or methanol/water extraction result in *trans*-rosmarinic acid as the most abundant phenolic acid [24,25,26]. The main components of the ethanolic extract obtained by Boufadi et al. are the phenolic acids rosmarinic-, salvianolic-, ferulic-, caffeic-, carnosic-, cinnamic- and sagerinic acid in order of decreasing quantity [27]. The flavonoids and phytochemicals catechin, cirsimaritin, quercetin, luteolin, hesperidine, apigenin and thymol were also detected in the ethanolic Sage extract [27]. These results confirm that Sage is rich in important health-promoting phytochemicals. The composition is widely dependent on extraction methods, climate, soil, plant age and irrigation patterns [21]. As a general result, rosmarinic acid is available in all extractions mentioned in the literature. Further components are caffeic- and ferulic acid, as well as further phenolic acids and flavonoids in decreasing amounts. A recent study by Boufadi et al. describes the maceration of 80 g fresh Sage leaves in 100 mL of pure ethanol in a sealed glass bottle for 72 h in darkness [27]. The product was filtered and a rotary evaporator was employed at 40 °C to obtain the pure extract [27]. We extracted 100 mg fresh Sage in 100 mL pure ethanol during a maceration period of 30 days and expect similar composition. A recent review concluded a higher polyphenol content for harvesting Sage plants in summer, while winter harvest leads to higher flavonoid content [18]. Extraction with ethanol as a solvent at lower temperatures seems to be ideal for dried Sage leaves [18]. An increase in extraction temperature leads to evaporation of volatile components and lowers polyphenol content [18]. Longer extraction time resulted in an increase in carnosic components and rosmarinic acid [18]. Keeping these factors in mind, we chose ethanol as solvent and macerated the fresh Sage plants at ambient temperatures sealed in glass bottles for 30 days. Our analytical results confirmed the availability of the needed bioactive compounds and their antimicrobial activity against 10 pathogens.

### 2.1. Elemental Composition and Morphological Examination

#### Electron Microscope (SEM) and Energy-Dispersive X-ray Spectroscopic (EDS) Analysis

SEM and EDS analysis were employed to investigate composition and morphology of AV-PVP-Sage-I_2_ (Figure 1).

AV-PVP-Sage-I_2_ shows a homogenous, amorphous pattern in the SEM analysis (Figure 1a). Figure 1b reveals mainly carbon, followed by iodine, oxygen, chlorine, potassium and sodium in the sample. The last three atoms were also present in our previous compound AV-PVP-I_2_ [31]. This indicates similarities in composition and purity of the title compound [31]. The EDS reveals higher intensity peaks in the Sage complex between 0–5 keV, suggesting higher polyphenol content in AV-PVP-Sage-I_2_ due to additional Sage plant extracts (Figure 1b). The samples were all coated with gold.

The adsorption of the biohybrid AV-PVP-Sage-I_2_ on different medical tissues is relevant for its application as antimicrobial agent. We impregnated sutures with our title compound. The material was analyzed by SEM and EDS analysis to rectify the coating process, composition and morphology of the adsorbed AV-PVP-Sage-I_2_ (Figure 2).

The plain, braided surgical suture is shown in Figure 2a [31]. The suture in Figure 2b is coated by a thick, homogenous layer of AV-PVP-Sage-I_2_ (11 µg/mL). Figure 2c depicts the resulting EDS of the dip-coated suture. The biohybrid effectively coats the surgical suture and may enable its use in the prevention of surgical site infections.

### 2.2. Spectroscopical Characterization

#### 2.2.1. UV-Vis Spectroscopy

The UV-vis spectrometric results are in agreement with previous studies (Figure 3) [31,56,72].

The two biohybrids AV-PVP-Sage and AV-PVP-Sage-I_2_ absorb in the range between 200 to 450 nm (Figure 3). The iodinated formulation is dominated by absorption peaks of smart triiodides at 291 and 359 nm in compliance with our previous investigations (Figure 3a,b) [31,56,59,60,61,62,63,72].

The fresh and three-day-old samples of AV-PVP-Sage-I_2_ absorb at shorter wavelengths (Figure 3). This blue shift happens after iodination and implies removal of the chromophores and the conjugation system. Higher energy is needed to overcome the increased energy gap between the bonding and antibonding orbitals in the molecules absorbing between 215–255 nm. Iodine molecules seem to form halogen and hydrogen bonding with the available molecules in the formulation. Due to these interactions, conjugation systems, double bonds in C=C and carbonyl bonds are affected. These interactions lead to more encapsulation and coiling of the whole biomolecular structure, as well as a size decrease. The hypsochromic effect suggests better incorporation of biomolecules and iodine moieties into the PVP polymer backbone between 215–255 nm. I_2_-molecules show absorption peaks at 203, 227 and 460 nm [31]. The title formulation AV-PVP-Sage-I_2_ absorbs at 202, 205, 290 and 359 nm. According to previous reports, these absorption peaks originate from iodide ions (202 nm), molecular iodine (205 nm) and triiodide ions (290 and 359 nm) [31,47,52,53,54,55,56,57]. We can infer the hydrolysis of iodine resulting in triiodide ions [74]. The triiodide ions form hydrogen bonds with the C=O groups of PVP [31]. This results in the formation of PVP-I_3_^−^ complexes within the biohybrid:AV-Sage-PVP + [I-I^…..^I]^−^ ⇆ AV-Sage-PVP-[I-I-I]^−^(1)
with absorptions at 290, 305 and 359 nm in agreement with previous investigations [31,64]. The complexed triiodide ions are purely halogen bonded [I-I-I]-units within the polymer PVP-framework. Our previous studies on smart triiodides reveal the same absorption maxima of λ-max = 290 and 359 nm (Figure 3, Table 1) [31,47,52,53,54,55,56,57]. Such absorption maxima are indicating pure halogen bonding within smart triiodide units [31,56,61]. Smart triiodides proved to have increased antimicrobial activities [31,56,61]. Their exceptional stability enables a controlled release of molecular iodine from the triiodide units [31,56,61]. The release starts, when electrostatic interactions between the cell membrane of microorganisms lead to deformations of the complex, polymer compound [31,56,61]. The formula PVP-I_2_ can be used interchangeably with PVP-I_3_^−^ [31,47,64,65,66,67,68,69,70,71,72].

The UV-vis spectrum of AV-PVP-Sage-I_2_ shows absorption peaks of molecular iodine at around 205 nm [31,75,76]. The broad absorption from 204–220 nm is due to iodine moieties within the structure of AV-PVP-Sage-I_2_ (Table 1).

Addition of iodine results in a hypsochromic effect due to encapsulation and complexation of the iodine in form of triiodide ions into the PVP polymer backbone. The coiling of PVP decreases the availability of π-electrons, the chromophores cannot absorb and hydrogen bonding is increased [74].

After 30 min, the AV-PVP-Sage-I_2_ sample exhibits decreased absorption intensities compared to the fresh sample in the region between 200–215 nm (Figure 3a). This confirms the encapsulation of the iodine moieties into the biocomplex by hydrogen bonding and reduced conjugation systems of the related plant phytochemicals. The incorporation of iodine, iodide ions and triiodide ions into the biopolymer leads to changes in the complex.

The spectroscopic analysis of AV-PVP-Sage-I_2_ after 3 days reveals two different trends.

The region after 284 nm shows lower absorption in the three-day-old AV-PVP-Sage-I_2_ sample compared to the same fresh sample (Figure 3b, Table 1). This result suggests the continued encapsulation of the triiodide moieties by hydrogen bonding through electrostatic attraction to the carbonyl groups of PVP according to Equation (1) [31,47,64,65,66,67,68,69,70,71,72]. The continued uptake and incorporation of triiodide ions triggers release of iodine molecules and iodide ions:AV-Sage-PVP-[I-I] ⇆ AV-Sage-PVP + [I-I](2)
AV-Sage-PVP-[I]^−^ ⇆ AV-Sage-PVP + [I] ^−^(3)
[I-I] + [I]^−^ ⇆ [I-I^…..^I]^−^(4)
according to Equations (2) and (3). The release of iodine and iodide ions leads to formation of new triiodide ions in order to preserve the equilibrium and is witnessed by increased absorption intensities in the range between 200 to 283 nm (Figure 3b). The UV-vis spectrum shows also a bathochromic shift combined with the mentioned higher absorption intensities in the range between 200 to 283 nm (Figure 3a, Table 1). Such circumstances suggest changes in the PVP structure from single bonded C–O back to C=O groups. Increasing availability of conjugated systems and chromophores can be expected due to Sage and AV compounds. This implies partial release of aloin, iodine and iodide ions from the PVP backbone (Figure 3a, Table 1).

In conclusion, we can verify the continued incorporation of triiodide moieties into the complex AV-PVP-Sage-I_2_ within a time frame of three days under storage in the fridge. This encapsulation of triiodide units into the biopolymer triggers a release of iodine and triiodide ions, as well as other AV-Sage phytochemicals during the process.

The UV-vis spectrum of AV-PVP-Sage shows strong, broad absorbance peaks with at λ-max = 283 and 338 nm (Figure 3, Table 1). These two absorptions are related to Sage and AV phenolic compounds in agreement with previous reports [14,20,24,31]. The spectroscopic analysis of AV, Sage and their encapsulation with PVP is provided in Figure 4.

Aloin, aloe-emodin, aloesin, 10-O-β-d-glucopyranosyl-aloenin, rhein, pyrogallol and hesperidin reveal absorption peaks between 200–360 nm (Figure 4a) [31]. Aloin is showing strong absorbance intensities at 206, 268, 296 and 353 nm with accordance with our previous study on AV-PVP-biohybrids (Figure 4a) [31]. The bands of aloin are strong and overlap with the other AV components (Figure 4a) [31]. The broad bands between 270–320 nm and 330–400 nm suggest the availability of more AV biomolecules (Figure 4a) [31]. These include aloe-emodin, which can be detected usually at 226, 257 and 285 nm [31]. Aloesin is available and can be confirmed by the absorptions at around 251, 296 and 360 nm (Figure 4a) [31]. Pyrogallol is confirmed by the absorption bands at around 280 and 349 nm (Figure 4a) [31]. Rhein absorbs at 329 and 295 nm and is available according to the UV-vis spectrum in Figure 4a [31]. The absorption peaks at 202 and 293 nm confirm the availability of 10-O-β-d-glucopyranosyl-aloenin in the AV sample (Figure 4a) [31]. Hesperidin exhibits an absorption signal at 329 nm in previous reports and can be detected here around 340 nm [31,41]. After adding PVP into the AV sample, the absorption intensities decrease dramatically as seen in Figure 4b. All the AV biomolecules are encapsulated into the PVP backbone. The chromophores cannot absorb and are fixed by hydrogen bonding or new bonds between the PVP carbonyl groups and other entities within the AV biocomponents. The inset in Figure 4b illustrates that the absorption bands of the concerned biomolecules appear at the same wavelengths but extremely lower absorption intensities.

The Sage extract contains compounds, which absorb also in the same range of AV biomolecules [20,24,28]. The addition of PVP leads to incorporation of Sage biomolecules into the PVP polymer matrix verified by the hypochromic shift towards lower intensities (Figure 4d). The pure Sage sample reveals a high intensity absorption band around 200–230 nm, followed by a broad shoulder around 230–50 nm and another a broad band from 260 to 300 nm (Figure 4c). Lower intensity absorption bands appear finally at 310–480 nm and 400–450 nm (Figure 4c). According to previous studies these bands can be assigned to Sage biomolecules [24,28].

The peak at 283 nm in pure Sage (Figure 4c) and in the formulation AV-PVP-Sage (Figure 3b) is related to sagerinic acid and salvianolic acid B [24]. In previous reports, these two compounds show absorption signals at 284 and 284/340 nm, respectively. Rosmarinic acid shows absorptions peaks at 280 nm and 330 nm [28]. Another study reports caffeic acid and *trans*-rosmarinic acid absorptions at 328 nm, while sagecoumarin absorbs at 332 nm [24]. The same absorption peaks appear in our formulations, as well as in the pure Sage sample, thus confirming the presence of *trans*-rosmarinic and caffeic acid (Figure 3b, Figure 4c). Boufadi et al. revealed in a recent publication the result of their ethanolic maceration extract [27]. Their extraction method is comparable to our method. We refrained from using heat in order to preserve all phytochemicals. They reported in their UV-vis analysis absorption signals at 270 nm for rosmarinic-, salvianolic-, ferulic-, caffeic-, carnosic-, cinnamic- and sagerinic acid in decreasing order [27]. They analyzed the sample at 320 nm and recorded the flavonoids and phytochemicals catechin, cirsimaritin, kaempferol, quercetin, luteolin, hesperidine, apigenin and thymol, among others [27]. Our compound AV-PVP-Sage reveals a peak at 331 nm, which proves the availability of sagecoumarin, caffeic acid and trans-rosmarinic acid in our extract (Figure 3b). The same absorption bands appear in Figure 4c,d.

AV-PVP-Sage exhibits generally lower absorbance intensity in comparison to AV-PVP-Sage-I_2_ (Figure 3, Table 1). Higher absorbance intensity and a red shift are recorded exceptionally between the wavelengths of 215–255 nm (Figure 3). As a result, there is higher availability of biomolecules consisting of chromophores with conjugated π-systems in this region. These are π-electrons from double bonds, aromatic ring systems or oxygen atoms with nonbonding orbitals originating from lone electron pairs. Such molecules in AV-PVP-Sage form less hydrogen bonding and therefore are less encapsulated between the wavelengths of 215–255 nm (Figure 3, Table 1). The related absorption wavelengths reveal PVP (221 nm), rhein (229), aloin (231 nm), aloesin (251 nm) and aloe-emodin (226 and 257 nm) as the reason for the combined bathochromic and hypsochromic shifts in the spectrum [31].

The sample was sealed and kept for three days in the fridge in darkness. The three-day-old AV-PVP-Sage reveals absorption intensities around 200–240 nm higher than the fresh sample. Under these conditions, the related molecules increased the conjugation in their aromatic ring systems and changed C-O to C=O bonds. Hydrogen bonding within the older sample was reduced. These results suggest changes in the molecular structure of the associated molecules by breaking intramolecular covalent bonding and dipol-dipol attraction (hydrogen bonding). We explain this by the partial release of rhein, aloin, aloesin and aloe-emodin from the PVP-polymer backbone in the biohybrid AV-PVP-Sage. In the region after 284 nm, the three-day-old sample exhibits lower absorption intensities and therefore better incorporation of the biomolecules into the biopolymeric structure. The concerned phytochemicals are sagerinic acid (284 nm), salvianolic acid (282 and 331 nm), trans rosmarinic acid (328 nm) and sage coumarine (332 nm).

AV and Sage components cannot be observed in the absorption spectra of AV-PVP-Sage-I_2_ due to the overlap with the strong bands of the triiodide ions at λ-max = 290 and 359 nm (Figure 3, Table 1). Addition of iodine leads to a hypochromic effect with lower absorbance for the peaks at 289 and 331 nm. The latter reveals stronger decrease in intensity and proves the encapsulation of the related Sage and AV components into the polymeric backbone. These compounds are caffeic acid, the caffeic acid trimer sagecoumarin, *trans*-rosmarinic acid and the AV related flavanone glycoside hesperidin.

AV-PVP-I_2_ absorbs with high intensity at 222 nm [31]. Adding the Sage extract resulted in a blue shift indicating a successful incorporation of the Sage extract into the polymer. This has an impact on the polymer backbone mobility by influencing local structures and is confirmed in the further spectroscopic analysis of the Sage samples. The Sage extract has strong interactions with the polymeric structure and reduces free movement of the CH groups. PVP chains are no more able to move freely due to Sage phenolic compounds [74]. Absorption peaks related to AV cannot be assigned clearly due to the strong overlap.

In conclusion, the presence of the AV biomolecules aloin, aloe-emodin, aloesin, 10-O-β-d-glucopyranosyl-aloenin, rhein, pyrogallol and hesperidin within both title compounds is confirmed (Figure 3 and Figure 4) [31]. The Sage components trans-rosmarinic-, salvianolic-, ferulic-, caffeic- and sagerinic acid, as well as sagecoumarin and hesperidin are also available in our two formulations (Figure 3 and Figure 4).

#### 2.2.2. Surface-Enhanced Raman Spectroscopy (SERS)

Raman spectroscopic analysis of AV-PVP-Sage-I_2_ is shown in (Figure 5).

The Raman spectrum reveals several high intensity peaks in the range between 100 to 400 cm^−1^. The highest peak is due to symmetrical vibration ν_1__s_ within I_3_^−^ units at 110 cm^−1^ (Figure 4). The medium-sized peak of the vibrational stretching mode ν_3__s_ at 144 cm^−1^ is accompanied by the weak and very weak asymmetric stretching modes ν_as_ at 222 and 334 cm^−1^, respectively (Figure 5, Table 2). The first can be considered an overtone of the I_3_^−^ band [77]. The triiodide moieties clearly dominate the absorption pattern in the spectrum (Figure 5, Table 2).

Triiodide bending vibration modes ν_2_ in form of very weak shoulders are available at 61 and 70 cm^−1^ [77]. The latter can be considered a hot band transition related to ν_2_ [77].

Iodine units show a very weak shoulder and a strong peak at 80 and 169 cm^−1^, respectively (Figure 5, Table 2) [77]. The first peak at 80 cm^−1^ is related to stretching vibrations in I_2_^….^I^−^ units [77]. The strong absorption signal at 169 cm^−1^ originates from I**^…^**I stretching vibrations, which are connected to a nearby I^−^ ion in form I**^…^**I^….^I^−^ [77]. The presence of the Raman active vibrational modes at 61, 85 and 169 cm^−1^ is due to non-linearity within the triiodide ion. The triiodide units in AV-PVP-Sage-I_2_ are verified here as unsymmetric and slightly non-linear in their free form before complexation (equation 1 and 4) [77]. The very weak absorbance by iodine moieties in the UV-vis spectroscopic analysis confirms the same results (Figure 5, Table 2). Pentaiodide units are not available in our title compound because they usually appear around 137 to 147 cm^−1^ and 167 cm^−1^ [59,78,79]. Free iodine absorbs usually at 172 cm^−1^, but is not available in free form in our biohybrid AV-PVP-Sage-I_2_ (Figure 5, Table 2) [53,59,80,81].

The phenolic acids caffeic acid, rosmarinic acid, ferrulic acid and vanillic acid are major components in the ethanolic Sage extracts [18]. AV contains cinnamic acid and acemannan next to other important phytochemicals [31]. The carboxylic acid absorptions of these plant components are necessarily part of the biohybrid AV-PVP-Sage-I_2_ [31]. The availability of these compounds are confirmed in the Raman spectrum by their weak asymmetric and very weak symmetric stretching -COOH absorptions at 1577 and 1457 cm^−1^, respectively [82]. Hydroxyl-groups show weak, broad bands at 3000 to 3600 cm^−1^ extracts [82]. In our Raman spectrum, these bands appear at lower wavelengths with very low intensity (2850 to 3000 cm^−1^) (Appendix A—Raman full). The weak band at 2942 cm^−1^ and the very weak band at 2939 cm^−1^ can be assigned to OH- and asymmetric CH_2_- stretching modes, respectively. The OH-bands show a red shift to lower wavelengths in AV-PVP-Sage-I_2_ compared to pure AV. The OH-stretching modes in our compound contain weaker C-OH bonds than in pure AV due to encapsulation and hydrogen bonding. The acetylation degree of the carboxyl groups in acemannan is usually evident with the absorption 1740 cm^−1^ [31]. Very weak asymmetric and symmetric C=O stretching bands of the carboxyl groups appear at 1708 and 1457 cm^−1^ in our title compounds, respectively (Appendix A–Raman full). O–C–O stretching vibrations are available at 1664 cm^−1^ (Appendix A–Raman full). Acemannan acetyl groups show very weak O–C–O absorptions at 1232 cm^−1^ (Appendix A–Raman full) [31]. Glucoside units H–C–O–H within acemannan absorb at 1232 cm^−1^ (Appendix A–Raman full) [31].

The weak absorption of these groups is overshadowed by the strong absorption of the iodide moieties within AV-PVP-Sage-I_2_ and makes further clarifications impossible.

#### 2.2.3. Fourier-Transform Infrared (FTIR) Spectroscopy

The FTIR analysis of the two samples confirmed the expected composition and mechanisms of complexation during the addition of molecular iodine (Figure 6, Appendix A).

The analysis reveals the same pattern like in our previous investigation of the AV-PVP-I_2_ biohybrid (Figure 5) [31]. The compounds AV-PVP-Sage and AV-PVP-Sage-I_2_ absorb at the same wavelengths (Figure 6a,b).

AV-PVP-Sage-I_2_ shows slightly higher peak intensities than AV-PVP-Sage in the absorption spectrum (Figure 6a,b). An increase in absorbance is seen for –COOH (3228.84 cm^−1^), –C=O (3000, 1680, 1649.14 and 1641.42 cm^−1^), –CH/–CH_2_/–CH_3_ (2937.59, 2900.94 cm^−1^), and C–O (1049.28 cm^−1^) groups (Figure 6a,b). This indicates higher availability of chromophores, higher absorption of functional groups and less hydrogen bonding. The encapsulation of the chromophores with above functional groups decreases by reduced hydrogen bonding in the iodinated compound [31].

During the iodination, the molecular iodine is reduced to iodide by the phytochemicals within AV and Sage [20]. The hydroxyl-functional groups of the latter undergo oxidation to carbonyl-groups. The phenolic acids and phytochemicals, which were encapsulated into the PVP matrix are partly released due to the addition of iodine into the formulation. Iodine is reduced to iodide ions, forms triiodide ions according to Equation (4), which partly replace the encapsulated larger phytochemicals in the PVP matrix in form of smart triiodides. These triiodide ions form hydrogen bonding to the C–O bonds in the PVP. The detached phenolic acids and polyphenols result in elevated numbers of functional groups –COOH, –OH, –C=O and –C–O. These lead to the above-mentioned higher absorption intensities in the iodinated formulation. At the same time, reduction of iodine molecules possibly triggers oxidation of hydroxyl groups to –C=O within phenolic acids and other phytochemicals originating from AV-Sage. This would explain the increase in absorption intensities for carbonyl groups. In the UV-vis spectrum, the triiodide ion absorption overshadows the weaker bands of AV and Sage biomolecules and prevents exact conclusions about their whereabouts.

The absorption band in the range between 3000 and 3600 cm^−1^ is higher and broader in our title compounds compared to pure PVP [31,74,83]. Both biohybrids contain phytochemicals with -OH and –COOH groups originating from AV and Sage extracts (Figure 6) [31,84].

The stretching vibration of –C=O groups in PVPI appears at 1650 cm^−1^ as a strong sharp peak (Appendix A). The FTIR spectra of our title compounds also show strong, broad absorptions with high peak intensities between 1800–1400 cm^−1^ (Figure 6). Similar broad bands are available in the FTIR spectrum of AV (Appendix A). The increased, broadened absorption bands are due to dipol-dipol interactions between PVP carbonyl groups, AV-PVP biomolecules and iodine moieties.

Methylene groups in pure PVP absorb in the region around 2700–2900 cm^−1^ were higher than in our compounds [31,74,83]. CH_2_-CH_2_ and C-C bond vibrations with smaller absorption intensities combined with a slight red shift surface after AV-PVP-Sage (887.26 cm^−1^, Figure 6a) was iodinated (885.33 cm^−1^ in Figure 6b). The red shift is an indicator for the weakening of the C-C bonds due to increased encapsulation of triiodide ions and the subsequent, partial release of AV-Sage biomolecules. At the same time, the absorption intensities at 1680, 1649.14 and 1641.42 cm^−1^ increase slightly after iodination of AV-PVP-Sage (Figure 6).

The absorption bands around 2940 and 2140 cm^−1^ originate from CH_2_-deformation and C=O combination vibration within PVP, respectively (Figure 6). Acemannan and pure PVP have absorption peaks at 2950 and 2922 cm^−1^ for asymmetric and symmetric stretching vibrations of CH_2_ chains, respectively [31,74]. The two title compounds AV-PVP-Sage and AV-PVP-Sage-I_2_ absorb with high intensity at 2945.30/2902.87 cm^−1^ and 2937.59/2900.94 cm^−1^, respectively (Figure 6, Appendix A). The peaks of AV-PVP-Sage-I_2_ have slightly higher intensity and are absorbing at lower frequency. The iodination resulted in this region with more encapsulation of triiodides into the PVP and the release of the larger acemannan molecules. Other large entities originating from the plant biomolecules are also partly released due to iodination. Similarly, the large carbohydrates monopyranose, pyranose and mannose are replaced by iodine, which is verified by the absorption increase at 1049.28 cm^−1^ after iodination (Figure 6b). Absorption intensities increase due to their reduced deformation or participation in the complex reaction with polyiodide ions [20].

In conclusion, C-C bonds characterized by methylene-, methyl- and -CH_2_ stretchings show increased absorption signal intensities. This signifies less encapsulation and less deformation of C-C entities.

The Sage extract also interacts with the polymer strongly by reducing the free movement of the CH groups [74]. The PVP chains are not more able to move freely due to the phytochemicals in Sage and AV (Appendix A) [74]. In our title compounds, very strong stretching modes of -COOH and –COO^-^ indicate reduced hydrogen bonding character within our two biohybrids compared to pure PVP. The broad bands around 3230 cm^−1^ are due to the –COOH groups within the AV and Sage compounds (Figure 6, Appendix A,). AV contains acemannan and cinnamic acid [74]. The main components in alcoholic Sage extracts obtained by the maceration method are phenolic acids and flavonoids [18,19]. Phenolic acids are mainly ferulic acid, rosmarinic acid and caffeic acid [18]. The presence of all these compounds is authenticated in the FTIR absorption spectrum in both title compounds by the bands around 3224, 1649, 1680, 1481, 1460, 1385, 1325, 1275, 1087 and 1053 cm^−1^ (Figure 6, Appendix A). Acemannan acetylation degree is confirmed by the bands at 1680 and 1275 cm^−1^ (Figure 6, Appendix A) [31]. The presence of acemannan is further validated by the H-C-OH stretching vibration of glycosidic groups and the O-C-O acetyl stretching around 1385 and 1275 cm^−1^, respectively (Figure 6, Appendix A) [31]. The symmetric (1420 cm^−1^) stretching mode of carboxylates verify hydrogen bonding between carboxylate group and I_3_^-^ anions (Figure 6, Appendix A). This results in higher coordination between the polymeric material and changes intramolecular ordering patterns of the PVP polymer backbone structure [78]. The medium sized band at 1088 cm^−1^ is due to the secondary alcoholic groups and the C-O stretching ester groups in the AV and Sage compounds (Figure 6, Appendix A). The peak at 887 cm^−1^ refers to aromatic ring CH-CH stretching vibrations present in AV and Sage extracts and CH_2_- bending deformations in the PVP alkyl chain (Figure 6, Appendix A).

The bands between 1500 and 1200 cm^−1^ are very weak compared to pure PVP. The latter has CH deformation vibrations at 1373 cm^−1^ and CH_2_ wagging, as well as C-N stretching vibrations at 1286 cm^−1^ [74]. These absorptions are also significantly decreased in our two Sage complexes (Figure 6, Appendix A). The C-C and CH_2_ rocking appear at 1017 cm^−1^ for pure PVP. In our title compounds, these are represented by absorption peaks around 1017 cm^−1^ (Figure 6, Appendix A).

Compared to AV-PVP-I_2_, and pure AV, our title compounds show higher absorption bands around 2900–3700 cm^−1^, 1600 and 1000 cm^−1^ (Figure 6, Appendix A). Hydrogen bonding and triiodide moieties bridge between the PVP backbone and the polyphenols. This reduces the free movement of the PVP and the acemannan chains. This leads to a partial release of AV and Sage biomolecules verified by increased absorption intensities of free –COOH, -OH and phenolic groups. The increase in hydrogen bonding is also confirmed in the broadening and increase of absorption in the range between 3000–3600 cm^−1^ (Figure 6, Appendix A). The biocomplex AV-PVP-Sage-I_2_ is a tightly packed, amorphous polymer.

### 2.3. X-Ray Diffraction (XRD)

The biohybrid AV-PVP-Sage-I_2_ shows two main broad peaks in the XRD spectrum at 2θ *=* 14° and 27° (Figure 7).

The broad band from 2θ *=* 20°–35° is due to strong presence of Sage [14]. The peak at 14° is related to AV and is in agreement with our previous studies of AV-PVP biohybrids [31]. The shoulder at 12.5° corroborates PVP (Figure 7). Two peaks also appear in the XRD analysis of AV-PVP-I_2_ at 2θ *=* 10° and 19° [31]. Rahma et al. studied the XRD of pure PVP and reported broad peaks at 2θ *=* 12° and 20°, indicating amorphous phases (Table 3) [74]. AV demonstrates crystalline behavior in previous studies [31,85,86,87]. The XRD analysis of AV-PVP-Sage-I_2_ suggests semicrystalline morphology with amorphous phases.

According to a recent study, pure AV gel shows crystalline peaks at 21°, 23°, 27° and 29° [82]. After complexing AV and iodine with PVP, the peaks appear with lower intensity and a shift to 10° and 19° in AV-PVP-I_2_ [31]. The decrease in intensity confirms the deposition of iodine moieties on the PVP in form of triiodide anions [31,78]. When the Sage extract is added, the intensities further decrease, accompanied by a shift to 2Theta 14^o^ and 28^o^ (Figure 6, Table 3). The decreased peak intensities, together with the shift to higher 2Theta values and the broadening indicate stronger encapsulation of the AV and Sage phytochemicals by the polymeric PVP backbone in AV-PVP-Sage-I_2_ (Figure 6, Table 3). The broadened peaks at 28° and 42° verify encapsulation of iodine moieties by the polymer. The broadening of the peaks are due to overlap with the iodine diffraction peaks [79]. The same peaks were available in the XRD of our previously studied compound AV-PVP-I_2_-NaI at 28° and 40° [31]. Previous studies reported peaks for crystalline iodine in their XRD analysis at 2Theta 25^o^, 28^o^ and around 30^o^ [31,68,78,88]. Zhang et al. describe one broadened iodine peak in the XRD analysis of their iodinated polymeric compound at 2θ *=* 22° [79].

The XRD analysis confirms the purity of AV-PVP-Sage-I_2_ due to missing of other phases (Figure 7, Table 3).

### 2.4. Determination of Antimicrobial Activities of AV-PVP-Sage and AV-PVP-Sage-I_2_

The antimicrobial properties of our two title compounds were determined by disc diffusion assay against 10 microorganisms. The tested Gram-positive strains were *S. pneumonia* ATCC 49619, *S. aureus* ATCC 25923, *S. pyogenes* ATCC 19615, *E. faecalis* ATCC 29212 and *B. subtilis* WDCM0003. The utilized Gram-negative bacteria were *E. coli* WDCM 00013 Vitroids, *P. mirabilis ATCC 29906*, *P. aeruginosa* WDCM 00026 Vitroids and *K. pneumonia* WDCM00097 Vitroids. The fungus *C. albicans* WDCM 00054 Vitroids was also tested against our biocompounds. Gentamicin was used as positive control antibiotics in comparison to the antimicrobial activity of AV-PVP-Sage and AV-PVP-Sage-I_2_. The negative controls ethanol and water showed no zone of inhibition (ZOI) and were not mentioned in the following Table 4.

The fungal reference strain *C. albicans* WDCM 00054 is much stronger inhibited by our biohybrid AV-PVP-Sage-I_2_ (ZOI = 52, 11 µg/mL) than the antibiotic nystatin (ZOI = 16) and AV-PVP-I_2_ (ZOI = 20, 25 µg/mL) (Table 4, Figure 8a) [31].

The Gram-positive pathogens strains *S. aureus* ATCC 25923 (Figure 8b) and *E. faecalis* ATCC 29212 show inhibitions zones of 20 and 15 mm, respectively. The same pathogens revealed against AV-PVP-I_2_ ZOI = 20 and 12 mm with a concentration of 25 µg/mL [31]. Brezoiu et al. studied a Sage extract embedded in silica and titanium mesoporous nanoparticles [21]. They reported a ZOI of 19 mm for the same *S. aureus* reference strain used in our study [21]. The susceptibility of *S. pneumoniae* ATCC 49619 and *S. pyogenes* ATCC 19615 towards AV-PVP-Sage-I_2_ is the same (ZOI = 14 mm) at the concentration of 11 µg/mL. The results of Brezoiu et al. with the same reference strains of *S. pneumoniae* and *S. pyogenes* exhibit inhibitions zones of 14 mm [21]. Our results appear in the same range without embedding into nanoparticles and at a lower concentration due to adding iodine [21]. AV-PVP-I_2_ inhibited these two Gram-positive microorganisms at a higher concentration of 25 µg/mL with by ZOI = 12 and 10 mm, respectively [31]. The spore forming microorganism *B. subtilis* WDCM 0003 is intermediately inhibited with ZOI = 13 mm by AV-PVP-Sage-I_2_. In comparison, AV-PVP-I_2_ achieved 17 mm inhibition zone with a concentration of 25 µg/mL [31]. The Gram-negative bacteria *K. pneumoniae* WDCM 00097 (Figure 8c) and *E. coli* WDCM 00013 show inhibition zones of 13 and 11 mm, respectively (Table 4). The same two Gram-positive pathogens exhibited ZOI of 14 and 13 mm towards AV-PVP-I_2_ at the higher concentration of 25 µg/mL [31]. A recent study mentions supercritical fluid extraction of Sage leaves and determination of antibacterial activities against clinical strains [89]. The Sage extract inhibited *B. subtilis, S. aureus, P. aeruginosa* and *E. coli* in descending order [89]. The authors suggest, that the antimicrobial compounds in their extract are mainly carnosic acid, rosmanol, ferruginol, rosmarinic acid and salvianolic acid [89]. Our biomaterial shows also stronger biocidal effects against Gram-positive bacteria compared to Gram-negative bacteria. This is an indicator for the availability of the compounds carnosic acid, rosmanol, ferruginol, rosmarinic acid and salvianolic acid [22]. Another study reveals low inhibition of 4 different *C. albicans* strains by the dried Sage leaves extracts of methanol/water, aqueous infusion and decoction methods [24]. *C. albicans* WDCM 00054 shows high susceptibility towards our title compound obtained by ethanolic maceration and addition of iodine.

In conclusion, all the studied microorganisms are susceptible towards our biomaterial AV-PVP-Sage-I_2_ except the two motile Gram-negative bacteria *P. mirabilis* ATCC 29906 and *P. aeruginosa* WDCM 00026. AV-PVP-Sage-I_2_ has higher antimicrobial activity at a lower concentration of 17 µg/mL compared to AV-PVP-I_2_ has with 25 µg/mL [31]. The addition of the Sage extract increased the inhibitory effect of the biohybrid at even lower concentrations of iodine. *P. aeruginosa* has multi-drug resistant efflux pumps for its defence against antibacterial agents and removes toxic substances out of the bacterial cell [23]. It was resistant against both title compounds, which contain phenolic acids. According to the UV-vis analysis, phenolic acids are available in Sage extracts in the form of sagecoumarin, *trans*-rosmarinic-, caffeic-, sagerinic- and salvianolic acid B [23,24,25]. AV contains only contains salicylic acid and cinnamic acid [31]. These phenolic acids in our two biocomplexes could not inhibit *P. aeruginosa* by disabling its efflux pumps.

AV-PVP-Sage-I_2_ is a strong antifungal agent against *C. albicans*. The Gram-positive bacteria *S. aureus* is the strongest inhibited strain. The title compound displays intermediate antibacterial activity against the Gram-negative microorganisms *K. pneumoniae* WDCM 00097 and *E. coli* WDCM 00013 (Table 4, Figure 8c).

We impregnated braided surgical sutures with AV-PVP-Sage-I_2_ and performed antimicrobial test on the same 10 reference strains (Figure 9).

The antimicrobial studies of dip-coated sutures with the iodinated title compound exhibited the same trends like the disc diffusion studies. The fungus *C. albicans* showed the strongest susceptibility, followed by the Gram-positive pathogens *S. aureus*, *S. pneumonia*, *B. subtilis*, *E. faecalis*, and *S.pyogenes* (Figure 9, Table 4). The two Gram-negative bacteria *K. pneumonia* and *E. coli* were inhibited, while *P. mirabilis* and *P. aeruginosa* were resistant.

The inhibitory effects of the biohybrid AV-PVP-Sage-I_2_ exhibit an interesting pattern. The fungus is most susceptible to the title compound, followed by strong to intermediate inhibitory effects on Gram-positive and Gram-negative pathogens (Table 4). The multi-drug resistant species *P. aeruginosa* WDCM 00026 and the highly motile swarmer *P. mirabilis* ATCC 29906 with multiple flaggellae are resistant against AV-PVP-Sage-I_2_. The bacterial morphology is related to the susceptibility against our compound. The highest antimicrobial action is exhibited against round cocci, followed by rod shaped bacilli. Non-motile species are more vulnerable than motile bacteria. The higher the agglomeration, the higher is the susceptibility against our title compound. The Gram-positive species forming clusters (*S. aureus*), chains (*E. faecalis*, *S. pyogenes*) and pairs (*S. pneumoniae*) are more inhibited than single, motile Gram-negative microorganisms (Table 4). The most susceptible organism is *C. albicans* with a less complicated cell membrane, followed by Gram-positive bacteria. Both groups of pathogens have less negatively charged outer cell membranes than the highly negatively charged Gram-negative bacteria [18,23,31]. The latter have a complicated outer cell membrane compared to the two former [18,23,31]. Gram-positive pathogens have a thick peptidoglycan layer crosslinked by peptides [18,23,31]. The negatively charged peptidoglycan layer has inclusions called teichoic acid and lipoteichoic acid [31]. These inclusions and the crosslinking peptides contain partial negatively charged oxygen atoms [31]. These form dipol-dipol and electrostatic interactions with the AV and Sage phytochemicals. Such interactions decoil the encapsulating PVP polymeric backbone resulting in deformation and release of iodine molecules. Iodine molecules subsequently destabilize the outer membranes by iodination of the cell wall components which leads to cell membrane destruction and cell leakage [18,23,31]. Iodine molecules diffuse through the membranes into and inhibit all the important processes from within resulting in cell death [31].

## 3. Materials and Methods

### 3.1. Materials

Iodine (≥ 99.0%), polyvinylpyrrolidone (PVP-K-30), ethanol (analytical grade), Sabouraud Dextrose broth, and Mueller Hinton Broth (MHB) were obtained from Sigma Aldrich (St. Louis, MO, USA). Disposable sterilized Petri dishes with Mueller Hinton II agar, McFarland standard sets, gentamicin (9125, 30 µg/disc) and nystatin (9078, 100 IU/disc) were purchased from Liofilchem Diagnostici (Roseto degli Abruzzi (TE), Italy). The bacterial strains *E. coli* WDCM 00013 Vitroids, *P. aeruginosa* WDCM 00026 Vitroids, *K. pneumoniae* WDCM 00097 Vitroids, *C. albicans* WDCM 00054 Vitroids, and *Bacillus subtilis* WDCM 0003 Vitroids were received from Sigma-Aldrich Chemical Co. (St. Louis, MO, USA). *S. pneumoniae* ATCC 49619, *S. aureus* ATCC 25923, *E. faecalis* ATCC 29212, *S. pyogenes* ATCC 19615 *and*
*P. mirabilis* ATCC 29906 were purchased from Liofilchem (Roseto degli Abruzzi (TE), Italy). Sterile polyglycolic acid (PGA) surgical sutures (DAMACRYL, 75 cm, USP: 3-0, Metric:2, 19 mm, DC3K19) were purchased from General Medical Disposable (GMD), GMD Group A.S., Istanbul, Turkey. Sterile filter paper discs with a diameter of 6 mm were purchased from Himedia (Jaitala Nagpur, Maharashtra, India). *Aloe vera* leaves (*Aloe barbadensis* Miller, AV) were harvested from the botanical garden of Ajman University campus, Ajman, UAE. Ultrapure water was utilized. All reagents were of analytical grade and used as delivered.

### 3.2. Preparation of Aloe vera (AV) Extract and Sage Extract

We harvested the leaves of an *Aloe vera* (*Aloe barbadensis* Miller) plant in the first week of December 2020, between 8:45–9:15 am, from the botanical garden of Ajman University. The 3-year-old plant was growing under the shade of the trees and was immediately carried to the nearby research laboratory of the College of Pharmacy and Health Sciences. The AV leaves were between 36 to 51 cm long. They were washed with distilled water to remove sand, then rinsed once with pure ethanol, several times with ultrapure water, and finally dried carefully. We sliced the AV leaves with a sterile knife and scrapped the gel out into a blender. The mucilaginous, pure gel was stirred for 20 min at maximum speed and then centrifuged at 4000 rpm for 40 min (3K 30; Sigma Laborzentrifugen GmbH, Osterode am Harz, Germany). The light-yellow supernatant was stored in a sealed brown bottle in darkness at 3 °C for further use.

Fresh Sage plants from Lebanon were purchased in December from the local store, washed several times with distilled water and then twice with absolute ethanol. 100 mg of fresh Sage plant was cut in pieces and placed into a sterile glas bottle. 100 mL absolute ethanol was added and the glas bottle was sealed. The maceration process lasted for 30 days at ambient temperatures with 4 times daily shaking the bottle manually. After 30 days, the filtrate was filled into a brown, disinfected glas bottle, sealed and stored in darkness at 3 °C for further use.

### 3.3. Preparation of AV-PVP-Sage and AV-PVP-Sage-I_2_

The preparation of the stock solution AV-PVP is achieved by dissolving 1 g polyvinylpyrrolidone K-30 (PVP) in 10 mL distilled water under stirring at room temperature (RT). 2 mL pure AV gel is placed in another beaker. Under continuous stirring at RT, we added into this beaker 1 mL of the PVP solution.

AV-PVP-Sage is prepared by mixing 2 mL of Sage ethanolic extract into the previously prepared 3 mL stock solution AV-PVP under stirring at RT.

The final formulation AV-PVP-Sage-I_2_ is obtained by first dissolving 0.05 g of iodine in 3 mL ethanol under stirring at RT. 1 mL of the iodine solution is added under continuous stirring into AV-PVP-Sage. In the previous study of AV-PVP biopolymers, we added 3 mL of iodine solution dissolved in methanol [31].

### 3.4. Characterization of AV Complexes

The title biocomplexes were investigated by SEM/EDS, Surface-Enhanced Raman Spectroscopy (SERS), UV-vis, and FTIR, and x-ray diffraction (XRD). The composition of our two biohybrids were confirmed by the analytical methods.

#### 3.4.1. Scanning Electron Microscopy (SEM) and Energy-Dispersive X-Ray Spectroscopy (EDX)

AV-PVP-Sage-I_2_ was analyzed at 15 kV by scanning electron microscopy (SEM), model VEGA3 from Tescan (Brno, Czech Republic). The instrument was equipped with an energy-dispersive X-ray spectroscopy (EDS). One drop of AV-PVP-Sage-I_2_ was dispersed in distilled water and placed onto a carbon-coated copper grid. The sample was dried under ambient conditions and coated with a gold film by a Quorum Technology Mini Sputter Coater. The SEM was employed to examine the morphology of the sample. The elemental composition of the biohybrid was analyzed by EDS analysis.

#### 3.4.2. UV-Vis Spectrophotometry (UV-Vis)

The UV-vis analysis of AV-PVP-Sage and AV-PVP-Sage-I_2_ was done by a UV-Vis spectrophotometer model 2600i from Shimadzu (Kyoto, Japan), using the wavelength range from 195 to 800 nm.

#### 3.4.3. Surface-Enhanced Raman Spectroscopy (SERS)

Raman analysis of the formulation AV-PVP-Sage-I_2_ was performed at room temperature on a RENISHAW (Gloucestershire, UK) combined with an optical microscope. The excitation of the solid-state laser beam with the wavelength of 785 nm, was focused onto the sample through the 50× objective of a confocal microscope (Spot diameter—2micron). The sample solution was filled into a standard cuvette (1 cm × 1 cm) and placed into the pathway of the laser beam. A CCD-based monochromator covering a spectral range of 50–3400 cm^−1^ Collected the scattered light. Output power was 10%, the spectral resolution −1 cm^−1^ and the integration time was —300 s.

#### 3.4.4. Fourier-Transform Infrared Spectroscopy (FTIR)

The FTIR analysis of AV-PVP-Sage and AV-PVP-Sage-I_2_ was performed on a ATR IR spectrometer with a Diamond window (Shimadzu, Kyoto, Japan). The samples were freeze-dried and then analyzed between 400 to 4000 cm^−1^.

#### 3.4.5. X-Ray Diffraction (XRD)

AV-PVP-Sage and AV-PVP-Sage-I_2_ were analyzed by XRD (BRUKER, D8 Advance, Karlsruhe, Germany). Cu radiation with a wavelength of 1.54060, coupled Two Theta/Theta with time/step of 0.5 s and a step size of 0.03 were employed.

### 3.5. Bacterial Strains and Culturing

The antimicrobial testing was performed with reference microbial strains of *S. pneumoniae* ATCC 49619, *S. aureus* ATCC 25923, *E. faecalis* ATCC 29212, *S. pyogenes* ATCC 19615, *Bacillus subtilis* WDCM 0003 Vitroids, *P. mirabilis* ATCC 29906, *E. coli* WDCM 00013 Vitroids, *P. aeruginosa* WDCM 00026 Vitroids, *K. pneumoniae* WDCM 00097 Vitroids, and *C. albicans* WDCM 00054 Vitroids. The reference strains were kept at −20 °C until inoculation with MHB by adding the fresh microbes. The resulting suspensions were stored at 4 °C until further use.

### 3.6. Determination of Antimicrobial Properties of AV, AV-PVP-I_2_, AV-PVP-I_2_-NaI, and AV-PVP-NaI

We investigated the antibacterial activities of AV-PVP-Sage-I_2_ against nine reference bacterial strains in comparison to the antibiotic gentamicin. These are the Gram-positive *S. pneumoniae* ATCC 49619, *S. aureus* ATCC 25923, *S. pyogenes ATCC* 19615, *E. faecalis* ATCC 29212, and *B. subtilis* WDCM 00003. The Gram-negative bacteria included *P. mirabilis* ATCC 29906, *P. aeruginosa* WDCM 00026, *E. coli* WDCM 00013, and *K. pneumoniae* WDCM 00097. *C. albicans* WDCM 00054 was utilized to investigate the antifungal activities and compared with the antibiotic nystatin as positive control. The negative controls of the solvents ethanol and water showed no inhibition zones and were not mentioned further. The antimicrobial tests on discs and sutures were repeated three times. We reported the average of the independent experiments in this study.

#### 3.6.1. Procedure for Zone of Inhibition Plate Studies

We utilized the zone of inhibition plate method to investigate the antimicrobial activity of AV-PVP-Sage-I_2_ against the selected microorganisms [89]. The bacterial strains were suspended in 10 mL MHB and incubated at 37 °C for 2 to 4 h. The fungal strain *C. albicans* WDCM 00054 was incubated on Sabouraud Dextrose broth at 30 °C. We evenly seeded disposable, sterilized Petri dishes with MHA with microbial culture of 100 μL with sterile cotton swabs. The microbial culture was previously adjusted to 0.5 McFarland standard. The Petri dishes were dried for 10 min and utilized in the antimicrobial testing.

#### 3.6.2. Disc Diffusion Method

The antimicrobial testing was performed according to the Clinical and Laboratory Standards Institute (CLSI) recommendations against the antibiotic discs of gentamycin and nystatin [90]. We soaked 2 mL of AV-PVP-Sage-I_2_ with concentrations of 11 µg/mL, 5.5 µg/mL, 2.75, and 1.38 µg/mL on sterile filter paper discs for 24 h. After removing the discs from the solution, we dried the discs for 24 h under ambient conditions. *C. albicans* WDCM 00054 was incubated for 24 h at 30 °C on agar plates. We measured the diameter of the zone of inhibition (ZOI) with a ruler to the nearest millimeter. The microbial inhibition towards AV-PVP-Sage-I_2_ is revealed in the diameters of clear inhibition zone around the disc. The reference strains are considered resistant, if there is no inhibition zone.

### 3.7. Preparation and Analysis of Dip-Coated Sutures

AV-PVP-Sage-I_2_ was dip-coated on uncoated sterile, multifilamented PGA sutures of approximately 2.5 cm. These sutures treated with acetone, dried at room temperature and immersed for 18 h into 50 mL of AV-PVP-Sage-I_2_ solution (11 µg/mL) at 25 °C. The blue color of the sutures changed to brown-blue. The impregnated sutures were then dried for 24 h under ambient conditions. The dried sutures were tested in vitro by the zone of inhibition assay against the 10 reference strains (*S. pneumoniae* ATCC 49619, *S. aureus* ATCC 25923, *E. faecalis* ATCC 29212, *S. pyogenes* ATCC 19615, *Bacillus subtilis* WDCM 00003, *E. coli* WDCM 00013, *P. aeruginosa* WDCM 00026, *P. mirabilis* ATCC 29906, *K. pneumoniae* WDCM 00097, and *C. albicans* WDCM 00054). We compared the uncoated and the impregnated sutures by SEM analysis.

### 3.8. Statistical Analysis

We used SPSS software (version 17.0, SPSS Inc., Chicago, IL, USA) for the statistical analysis. The data is represented as mean. The statistical significance between groups is calculated by one-way ANOVA. Any value of *p* < 0.05 was considered statistically significant.

## 4. Conclusions

Easily accessible, one-pot, bio-synthesized biopolymers with iodine are potential alternatives to overcome increasing antimicrobial resistance. Our biohybrid is a strong antifungal agent against *C. albicans* even at a very low concentration of 2.75 µg/mL. The formulation exerted antibacterial activities against five Gram-positive and two Gram-negative pathogens. The Gram-negative pathogens *E. coli* and *K. pneumonia* were intermediately susceptible to our compound. The majority of the studied microorganisms are susceptible at a very low concentration of iodine. Such low concentrations may reduce the incidence of iodine related disadvantages. The iodine release allows longer lasting effects, in synergism with the available phytochemicals within AV and Sage. The results may enable the use of AV-PVP-Sage-I_2_ as a microbicidal disinfectant and coating agent against resistant microorganisms. The inhibitory action of AV-PVP-Sage-I_2_ on sutures follows the same trends and potentiates prevention of surgical site infections. Further investigations are planned to assess the suitability of the title formulation AV-PVP-Sage-I_2_ for coating wound dressing materials and personal protective equipment. Smart bio-antimicrobials are interesting alternatives as sustainable basic disinfectants and coating materials. Such biomaterials may serve the public during health care system failures.

## Figures and Tables

**Figure 1 molecules-26-03553-f001:**
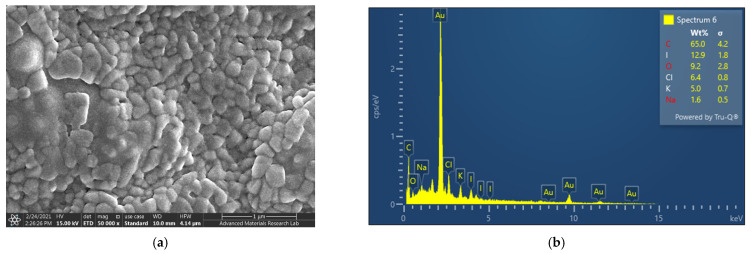
Scanning electron microscopy (SEM) (**a**) and Energy dispersive spectroscopy (EDS) (**b**) of AV-PVP-Sage-I_2_.

**Figure 2 molecules-26-03553-f002:**
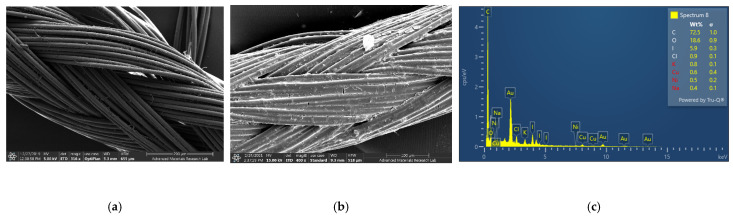
Scanning electron microscopy (SEM) of sutures: (**a**) plain suture [31]; (**b**) dip-coated with AV-PVP-Sage-I_2_; (**c**) Energy dispersive spectroscopy (EDS) of coated sutures.

**Figure 3 molecules-26-03553-f003:**
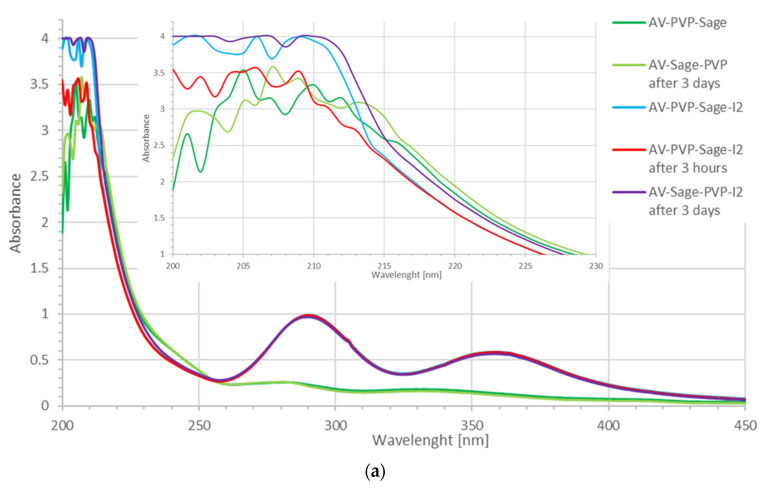
UV-vis analysis of AV-PVP-Sage and AV-PVP-Sage-I_2_: (**a**) 200–500 nm with inset of 200–230 nm; (**b**) 240–450 nm. (AV-PVP-Sage: dark green-fresh, light green-3 days old; AV-PVP-Sage-I_2_: blue-fresh, red after 3 h, purple-3 days old).

**Figure 4 molecules-26-03553-f004:**
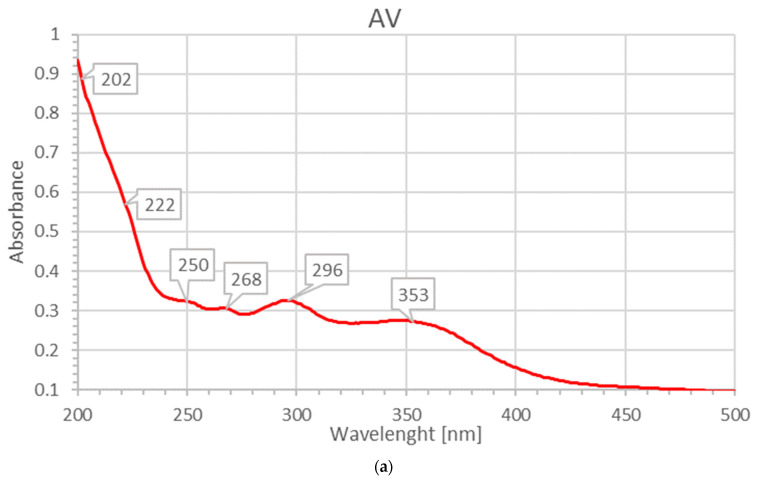
UV-vis analysis of pure plant extracts and their encapsulation: (**a**) AV; (**b**) AV-PVP; (**c**) Sage; (**d**) Sage-PVP.

**Figure 5 molecules-26-03553-f005:**
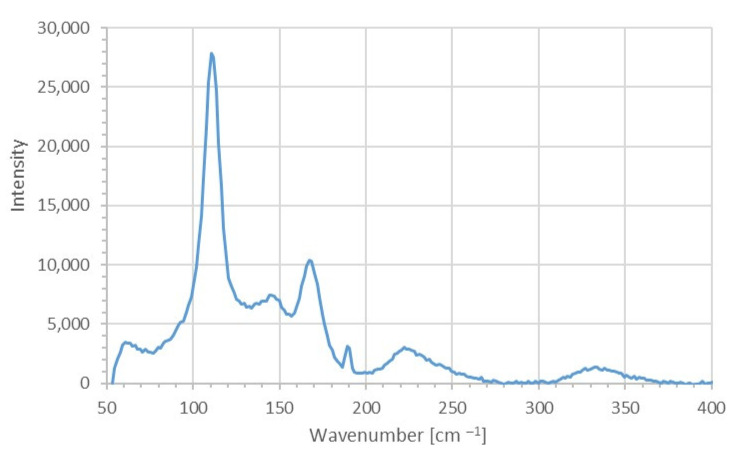
SERS spectroscopic analysis of AV-PVP-Sage-I_2_.

**Figure 6 molecules-26-03553-f006:**
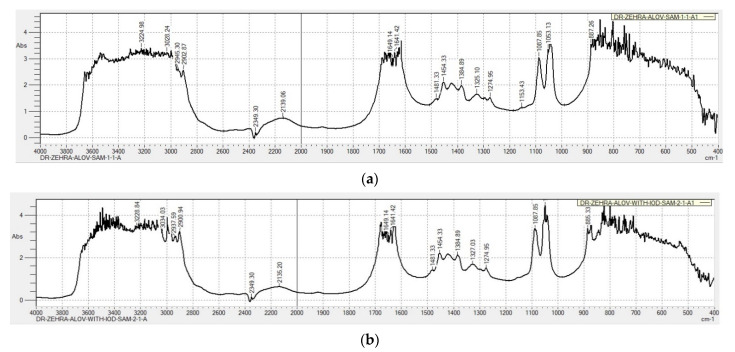
Fourier-Transform Infrared (FTIR) spectroscopic analysis of samples: (**a**) AV-Sage-PVP; (**b**) AV-PVP-Sage-I_2_.

**Figure 7 molecules-26-03553-f007:**
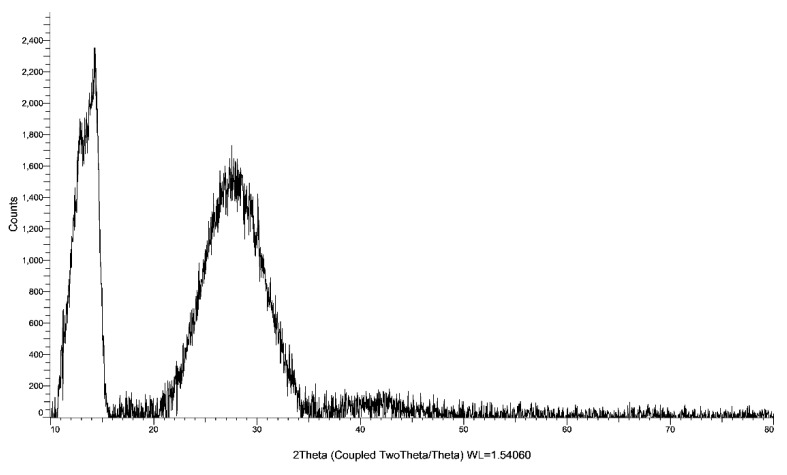
X-ray diffraction (XRD) analysis of AV-PVP-Sage-I_2_.

**Figure 8 molecules-26-03553-f008:**
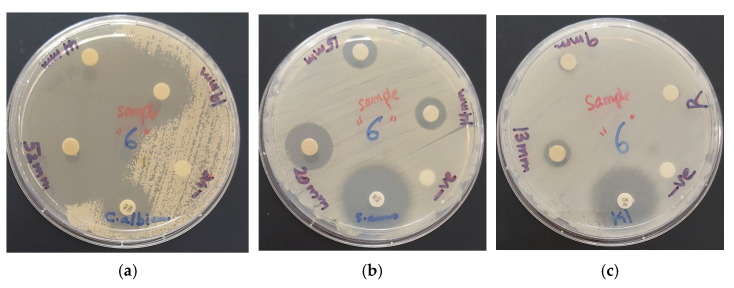
Disc diffusion assay of AV-PVP-Sage-I_2_ (at concentrations of 11, 5.5, 2.75 and 1.38 µg/mL) with positive control antibiotic gentamicin (30 µg/disc). From left to right: AV-PVP-Sage-I_2_ against (**a**) *C. albicans* WDCM 00054; (**b**) *S. aureus* ATCC 25932; (**c**) *K. pneumoniae* WDCM 00097.

**Figure 9 molecules-26-03553-f009:**
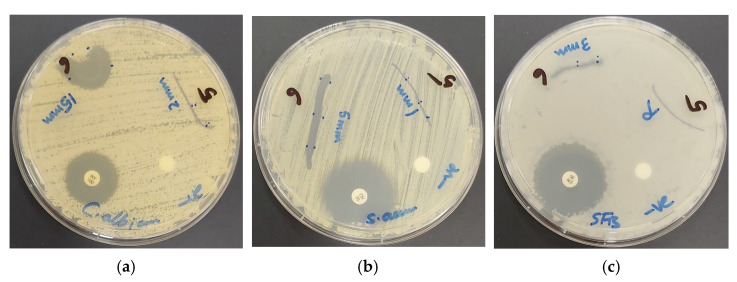
Dip-coated sutures with AV-PVP-Sage-I_2._ with positive control antibiotic gentamicin. From left to right: AV-PVP-Sage-I_2_ (11 µg/mL) against (**a**) *C. albicans* WDCM 00054; (**b**) *S. aureus* ATCC 25932; (**c**) *B. subtilis* WDCM 00003.

**Table 1 molecules-26-03553-t001:** UV-vis absorption signals in the samples AV-PVP-Sage (1), AV-PVP-Sage-I_2_ (2), AV-PVP-I_2_ (3), AV-PVP-I_2_-NaI (4) and previous reports (nm).

Group	1 *	1 * (3 Days)	2 *	2 *(3 Days)	3 *	4 *	[20]	[24]
I_2_			206 vs	206 vs	205	203		
I_3_^−^			290 s,br 359 s,br	290 s,br 359 s,br	290 358	290 359		
I^−^			202 vs	202 vs	202			
AV/Aloin	206 vs	209 vs	206 vs	207 vs				
PVP	201 s,br	202 vs	201 s,br	202 s,br	222	212		
	203 sh	205 sh	203 sh	203 s,br				
	205 vs	207 vs	205 vs	205 s,br				
	210 vs	213 vs	209 vs	210 s,br				
	211 br	214 br	211 br	212 s,br				
	216 sh	217 sh	215 sh	216 sh				
PVP-I_2_	305 s,sh	305 s,sh	305 s,sh	305 s,sh	305	305		
Sage	283 w,br	283 w,br	**	**			270, 280	284/340
	338 w,br	338 w,br	340 m,sh	340 m,sh			330	328, 332

* UV-vis absorption signals with concentration of 11 µg/mL. ** The broad band around 290 nm overlaps and several peaks related to AV and Sage compounds cannot be observed. vw = very weak, br = broad, s = strong, vs. = very strong, m = intermediate, sh = shoulder.

**Table 2 molecules-26-03553-t002:** Raman shifts in AV-PVP-Sage-I_2_ (1) (cm^−1^).

Group	1	[77]	[78]	[79]	[59]	[80]	[81]
I_2_	s169*ν _as_ I-I^….^I^−^ sh,w80* I_2_^….^I^-^	m160*ν_as_ I-I^….^I^−^ m80* I_2_^….^I^−^			s172 ν_s_	172 ν_s_	s172 ν_s_
I_3_^−^	w189*ν I_2_^….^I^−^ sh,w61 δ_def_ sh,w70ν_2bend_ vs110 ν_1,__s_ m144 ν_3,__as_ w222 2ν_1,as_ vw334 ν_as_	sh60 δ_def_ sh,w75ν_2bend_ vs110 ν_1,__s_ m144 ν_3__s_ m154 ν_3,__as_ w221 2ν_1,as_	114 ν_1__s_ 143 ν_3__s_	217 ν_as_ 331 ν_as_	w232ν_as_ vw351ν_as_		227ν_as_ 340ν_as_
I_5_^−^				147 ν_as_*	m142 ν_as_	137 ν_as_	


ν = vibrational stretching, s = symmetric, a = asymmetric, 1 = stretching mode 1, 3 = stretching mode 3, bend = bending, δ_def_ = deformation. * belong to the same asymmetric, nonlinear unit I_3_^−^ = I_2_^……^I^−^.

**Table 3 molecules-26-03553-t003:** XRD analysis of the samples AV-PVP-Sage-I_2_ (1), AV-PVP-I_2_ (2) [31], AV-PVP-I_2_-NaI (3) [31] and in previous reports (2Theta^o^).

Group	1	2	3	[88]	[14]
I_2_	-	-	28 m 40 w	25 29 36	-
	-	-	-	-	-
PVP	13 s	10 m 19 s,br	11 s,br 20 s,br	-	-
Sage	28 s,br 42 w,br	-	-	-	32 s,s 48 s,s 55 m,s 57 m,s 69 w,s
AV	14 s	14 s 21 s,br 22 s,br	-	--	
	-	-	-	-	-

w = weak, br = broad, s = strong, m = intermediate.

**Table 4 molecules-26-03553-t004:** Antimicrobial testing of antibiotics (A), AV-PVP-Sage-I_2_ by disc dilution studies (1,2,3) and suture (S). ZOI (mm) against microbial strains by diffusion assay.

Strain	Antibiotic	A	1 ^+^	2 ^+^	3 ^+^	S
*S. pneumoniae* ATCC 49619	G	18	14	11	10	3
*S. aureus* ATCC 25923	G	28	20	15	14	5
*S. pyogenes* ATCC 19615	G	25	14	12	10	2
*E. faecalis* ATCC 29212	G	25	15	12	10	2
*B. subtilis* WDCM 00003	G	21	13	12	11	3
*P. mirabilis* ATCC 29906	G	30	0	0	0	0
*P. aeruginosa* WDCM 00026	G	23	0	0	0	0
*E. coli* WDCM 00013	G	23	11	0	0	1
*K. pneumoniae* WDCM 00097	G	30	13	9	0	2
*C. albicans* WDCM 00054	NY	16	52	41	19 *	15

^+^ Disc diffusion studies (6 mm disc impregnated with 2 mL of 11 µg/mL (1), 2 mL of 5.5 µg/mL (2) and 2 mL of 2.75 µg/mL (3) of AV-PVP-Sage-I_2_. A = G Gentamicin (30 µg/disc). NY (Nystatin) (100 IU). S suture and M mask tissue dip-coated with 2 mL of 17 µg/mL AV-PVP-Sage-I_2_. Grey shaded area represents Gram-negative bacteria. 0 = Resistant. * Further dilution to 1.38 µg/mL yielded ZOI = 10 mm. No statistically significant differences (*p* > 0.05) between row-based values through Pearson correlation.

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
