# Peer review of "Facile Synthesis of Bio-Antimicrobials with “Smart” Triiodides"

_molecules, 2021, doi:10.3390/molecules26123553_

Round 1

Reviewer 1 Report

Searching for new antimicrobial agents is obviously urgent nowadays, therefore studies presented in the manuscript deserve attention. However, I find some weakness within it. Authors has claimed that "AV and Sage are natural products which may have variable composition according to their environmental factors during growth, harvesting and extraction methods [21,31,37,42-45]. These factors influence their biological activities and cannot be standardized as such." And this is very much true. Why authors didn't decide to use pure active substances of AV and Sage? It would remove the standarsation problem away. 
Otherwise it is impossible to use these formulations in medical applications.
My other concern is thet the authors call their products as 'compounds'. In my opinion they should use the word 'formulations' rather than 'compounds' since as they has wrote PVP is an encapsulating agent.
It would impriove the manuscript significantly if the authors added a release profiles of selected antimicrobial substances from PVP matrix.

Author Response

Thanks  a lot dear Reviewer. Your valuable comments helped us to improve our manuscript.

Reviewer 2 Report

The manuscript on AV-PVP-SAGE iodine hybrids contains more weaknesses than strengths. The current format is far from publication quality. The analytical results presented are from poorly designed experiments. Authors need to perform the missing experiment and, please, complete the manuscript with the results before resubmission.

- The manuscript attempts to provide information and confirmation for PVP-encapsulated Sage compounds and iodine, with moderate success. The manuscript's fundamental weakness is that the assumed encapsulation has been unconfirmed because of inadequate evidence. In general, when the main goal is to prove encapsulation, the authors need to record analytical (mainly spectral) data for all the participating components, the physical mixture (if possible), and the putative encapsulated substance. These are missing from this manuscript.

- Figure 3 shows only AV-PVP-SAGE and AV-PVP-SAGE iodine, while in lines 225-226, the authors attribute the hypsochromic effect to iodine. This effect may be, but without evidence, it is an idea only. The authors themselves wrote that the I2/I3- peak overlaps with their composite. A simple doping or concentration-dependent peak shift could have validated this idea. Possibly they did these experiments but failed to insert the spectra. Although the authors have prepared a table, that is not a substitute spectra. Seeing is believing.

- Figure 4 shows only one Raman spectrum. It is hard to believe that this figure contains any information. The missing I2, I3-, and AV-PVP-SAGE spectra are because of obliviousness, or those spectra disprove the authors' theory?

- Figure 5 shows the impurity on the instrument rather than an IR spectrum. The maximum absorption is <0.05 AU!  And if they are going to record all the missing spectra, feel free to use the instrument's software to make the corrections and get informative spectra. Additionally, section 4.4.4 is incomplete. How did the authors record the spectra? In KBr, neat, or the instrument is an ATR IR machine?

- Both textual and SI tabular IR analysis is a bad joke! You can't see anything from the inset IR, and besides the fact that the spectrum is a noise, defining locating peaks from a broad, sawed IR band to support encapsulation is out of the correct science. Also, a single peak in the H-bonding/aromatic range looks more like a spike than a peak. Negative peaks (just above 1000 cm-1 and below 3000 cm-1) are also hard to interpret. Are the authors serious in saying that in the 1550-1750 cm-1 region, the bell-like curve has distinct peaks at 1585, 1637, 1653 cm-1? Did they deconvolute the spectrum which allows them to write that? 
The IR assignment in SI is more like copy/paste from a spectroscopy book - if this is true, it would be advisable to cite it - than an analysis of the spectrum presented. 1500m, br, sh, and 1463 cm-1 C=O) are in the SI, however, there is virtually no peak at 1500 cm-1; 2) spectrum seems flat at 1463(!) cm-1; 3) the 2000-2400 cm-1 region is just noise.

- What is the meaning of XRD? Especially, because of the lacking individual components and physical mixture XRDs! Yes, the composite is an amorphous material, which without XRD was not obvious? Table 3 is also an unintelligible section.

- The footnote of Table 4 has a better location in the experimental section.

- The author's contribution section (lines 693-698) is complicated. The manuscript has two authors, and it seems both of them worked equally in all steps. "Both authors participated equally in the experiments, and manuscript preparation" - or something like that - would be enough.

- As far as the referee knows, the "et al." in the reference section is not allowed. Please, check all references for consistent format and typos. In some cases, doi is missing (like ref. 50).
References 90 and 9 are unclear.

Author Response

Thank you very, very much.

Sorry for all the inconvenience.

Best regards

Zehra

Reviewer 3 Report

Dear Editor,

The manuscript submitted by Zehra Edis et al. and entitled: “Antimicrobial Properties of PVP-Encapsulated Salvia officinalis L.-Aloe Vera Barbadensis Miller–Iodine Hybrids” is an interesting study which deals with synthesis of material based on aloe vera extract with potential biological activity and more especially as antimicrobial material. This manuscript can be accepted after major revisions. Please to see my comments below.

Comments:

  • Authors should reformulate the title of the revised manuscript. It seems too confuse for readers.
  • In introduction part, authors have made lot of statements on antimicrobial extract/molecules, and few informations of antimicrobial material without backed it up with last good references in this field. Please to add new references in the revised manuscript. And Reduce the size of introduction (it seems too long for a research paper).
  • In result part, the data given were not really explained and discussed. Authors just made a very shallow presentation of their results. Please update the discussion part in the revised manuscript.
  • In the conclusion part, the aspects of novelty and the biological applications should be more underlined.

General comment:

  • In the revised manuscript, the authors need to pay more attention to grammatical construction of sentences and spelling of sentences.
  • The quality of figures should be improved in the revised manuscript.

Author Response

Thank you for your interesting advices.

It helped improving the manuscript.

Best regards

Zehra

Round 2

Reviewer 2 Report

The authors have significantly improved the quality of the manuscript during the major revision. They corrected their paper by the main concerns of the reviewers, and now the paper is ready for publication.

They should consider two minor recommendations in future publications:

  • They need to record and smooth an oversampled UV spectrum to avoid spiky lines (i.e. record at a resolution of 0.15-0.3 nm despite the 1-2 nm resolution of the instrument).
  • Another note is that below ~220 nm, the peaks are of little significance, and try to avoid peak picking in that region, as in Figure 4c-d.

These comments do not affect the actual content of this manuscript.

Reviewer 3 Report

Authors revised the manuscript according to reviewer's comment.

Consequently, paper could be accepted in present form.